# Comparison of the Transcriptomic Signatures in Pediatric and Adult CML

**DOI:** 10.3390/cancers13246263

**Published:** 2021-12-14

**Authors:** Minyoung Youn, Stephanie M. Smith, Alex Gia Lee, Hee-Don Chae, Elizabeth Spiteri, Jason Erdmann, Ilana Galperin, Lara Murphy Jones, Michele Donato, Parveen Abidi, Henrique Bittencourt, Norman Lacayo, Gary Dahl, Catherine Aftandilian, Kara L. Davis, Jairo A. Matthews, Steven M. Kornblau, Min Huang, Nathan Sumarsono, Michele S. Redell, Cecilia H. Fu, I-Ming Chen, Todd A. Alonzo, Elizabeth Eklund, Jason Gotlib, Purvesh Khatri, E. Alejandro Sweet-Cordero, Nobuko Hijiya, Kathleen M. Sakamoto

**Affiliations:** 1Department of Pediatrics, Stanford University School of Medicine, Stanford, CA 94305, USA; minyoung@stanford.edu (M.Y.); smithsm@stanford.edu (S.M.S.); heedon.chae@gmail.com (H.-D.C.); lcmurph@stanford.edu (L.M.J.); lacayon@stanford.edu (N.L.); mngdh@stanford.edu (G.D.); aftandil@stanford.edu (C.A.); kardavis@stanford.edu (K.L.D.); minhuang@stanford.edu (M.H.); nathan.sumarsono@utsouthwestern.edu (N.S.); 2Department of Pediatrics, University of California, San Francisco, CA 94143, USA; alex.lee2@ucsf.edu (A.G.L.); alejandro.sweet-cordero@ucsf.edu (E.A.S.-C.); 3Department of Pathology, Stanford University School of Medicine, Stanford, CA 94305, USA; spiterie@stanford.edu; 4Cytogenetics Laboratory, Stanford Health Care, Stanford, CA 94304, USA; jerdmann@stanfordhealthcare.org (J.E.); igalperin@stanfordhealthcare.org (I.G.); 5Institute for Immunity, Transplantation and Infection, Stanford University, Stanford, CA 94305, USA; mdonato@stanford.edu (M.D.); pkhatri@stanford.edu (P.K.); 6Stanford Center for Biomedical Informatics Research, Stanford University, Stanford, CA 94305, USA; 7Division of Hematology, Department of Medicine, Stanford Cancer Institute, Stanford University School of Medicine, Stanford, CA 94305, USA; pabidi@stanford.edu (P.A.); jason.gotlib@stanford.edu (J.G.); 8Hematology-Oncology Division, Charles Bruneau Cancer Center, Centre Hospitalier Universitaire Sainte-Justine, Montreal, QC H3T 1C5, Canada; henrique.bittencourt.hsj@ssss.gouv.qc.ca; 9Department of Leukemia, The University of Texas M.D. Anderson Cancer Center, Houston, TX 77030, USA; jmatthew@mdanderson.org (J.A.M.); skornblau@mdanderson.org (S.M.K.); 10Division of Pediatric Hematology/Oncology, Baylor College of Medicine, Houston, TX 77030, USA; mlredell@txch.org; 11Division of Hematology/Oncology, Children’s Hospital Los Angeles, Los Angeles, CA 90027, USA; cfu@chla.usc.edu; 12Department of Pathology, University of New Mexico Comprehensive Cancer Center, Albuquerque, NM 87102, USA; ichen@salud.unm.edu; 13Department of Preventive Medicine, University of Southern California, Los Angeles, CA 90032, USA; talonzo@childrensoncologygroup.org; 14Feinberg School of Medicine, Northwestern University, Chicago, IL 60611, USA; e-eklund@northwestern.edu; 15Department of Pediatrics, Columbia University Irving Medical Center, New York, NY 10032, USA; nh2636@cumc.columbia.edu

**Keywords:** pediatric CML, CML CD34+ cells, RNA sequencing, transcriptome, Rho pathway

## Abstract

**Simple Summary:**

To investigate whether pediatric and adult chronic myeloid leukemia (CML) have unique molecular characteristics, we studied the transcriptomic signature of pediatric and adult CML cells using high-throughput RNA sequencing. We identified differentially expressed genes and pathways unique to pediatric CML cells compared to adult CML cells. The Rho pathway was significantly dysregulated in pediatric CML cells compared to adult CML cells, suggesting the potential importance in the pathogenesis of pediatric CML. Our study is the first to compare transcriptome profiles of CML across different age groups. A better understanding of the biology of CML across different ages may inform future treatment approaches.

**Abstract:**

Children with chronic myeloid leukemia (CML) tend to present with higher white blood counts and larger spleens than adults with CML, suggesting that the biology of pediatric and adult CML may differ. To investigate whether pediatric and adult CML have unique molecular characteristics, we studied the transcriptomic signature of pediatric and adult CML CD34+ cells and healthy pediatric and adult CD34+ control cells. Using high-throughput RNA sequencing, we found 567 genes (207 up- and 360 downregulated) differentially expressed in pediatric CML CD34+ cells compared to pediatric healthy CD34+ cells. Directly comparing pediatric and adult CML CD34+ cells, 398 genes (258 up- and 140 downregulated), including many in the Rho pathway, were differentially expressed in pediatric CML CD34+ cells. Using RT-qPCR to verify differentially expressed genes, VAV2 and ARHGAP27 were significantly upregulated in adult CML CD34+ cells compared to pediatric CML CD34+ cells. NCF1, CYBB, and S100A8 were upregulated in adult CML CD34+ cells but not in pediatric CML CD34+ cells, compared to healthy controls. In contrast, DLC1 was significantly upregulated in pediatric CML CD34+ cells but not in adult CML CD34+ cells, compared to healthy controls. These results demonstrate unique molecular characteristics of pediatric CML, such as dysregulation of the Rho pathway, which may contribute to clinical differences between pediatric and adult patients.

## 1. Introduction

Chronic myeloid leukemia (CML) accounts for 2–9% of leukemias in children and adolescents and occurs with much greater frequency in adults. Compared to adults, children with CML tend to present with higher white blood cell (WBC) counts and larger spleens, suggesting that the biology of pediatric CML differs from that of adult CML [1,2].

We hypothesize that the differences in the clinical features of pediatric and adult CML are due to unique molecular characteristics. To test this hypothesis, we compared transcriptomic signatures of pediatric and adult CML CD34+ cells and healthy control CD34+ cells by performing high-throughput RNA sequencing analysis.

In this study, we found that several genes in the Rho pathway were differentially expressed in pediatric CML CD34+ cells compared to adult CML CD34+ cells. Our study is the first to compare transcriptome profiles of CML across different age groups.

## 2. Materials and Methods

### 2.1. Patient Samples and Clinical Data Analysis

Clinical and demographic features at diagnosis were extracted from electronic medical records or provided by the Children’s Oncology Group (COG) for pediatric (<18 years) and adult CML patients and compared using Fisher’s exact test (categorical variables) or Wilcoxon rank sum test (continuous variables). Bone marrow samples from CML patients were collected through voluntary participation in existing tissue banking studies at Stanford University School of Medicine, MD Anderson Cancer Center, and the COG in compliance with the institutional review boards. Informed consent for the tissue banking studies was obtained from all human subjects in accordance with the Declaration of Helsinki. For controls, bone marrow CD34+ cells from healthy bone marrow donors with a similar age range as the CML patients were provided by Université de Montréal for pediatric samples and purchased from StemCell Technologies (Vancouver, BC, Canada) for pediatric samples and from Lonza, Inc. (Basel, Switzerland) for pediatric and adult samples. Clinical and demographic features of individual CML patients and healthy controls are shown in Appendix A. Every effort was made to obtain complete clinical data; despite this, several historically treated patients had incomplete information available.

### 2.2. RNAseq Analysis

CD34+ cells were isolated by fluorescence-activated cell sorting (FACS) of pediatric CML (*n* = 9), adult CML (*n* = 10), pediatric healthy (*n* = 10), and adult healthy (*n* = 10) bone marrow samples. RNAseq was performed as previously published [3]. Briefly, prepared libraries were sequenced on Illumina HiSeq 4000 or Illumina NextSeq 500 instruments. Raw sequences were trimmed and aligned to the hg38 reference genome with STAR/2.5.1b aligner. Gene-level counts were determined with STAR-*quantMode* option using gene annotations from GENCODE (p5). Differential gene expression and pathway analysis were conducted with R/3.5.3. Counts were normalized with trimmed mean of M-values (TMM) from the EdgeR/3.24.3 package and further transformed with VOOM from the Limma/3.38.3 package. A linear model using the empirical Bayes analysis pipeline, also from Limma, was then used to obtain *p*-values, adjusted *p*-values, and log-fold changes (LogFC). We performed four pairwise comparisons: (1) pediatric CML vs. pediatric healthy, (2) adult CML vs. adult healthy, (3) pediatric CML vs. adult CML, and (4) pediatric healthy vs. adult healthy CD34+ cells. A false discovery rate (FDR) of ≤0.05 and absolute log2 fold-change > 1 was used to define differentially expressed genes (DEG) in each comparison. We additionally performed a single comparison of (pediatric CML vs. pediatric healthy) vs. (adult CML vs. adult healthy) with an FDR of ≤0.12, and an absolute log2 fold-change of >1 was used to define DEG. To identify potentially unique pathways based on DEG, pathway over-representation was calculated with either goana from the Limma package or clueGO, while Gene Set Enrichment Analysis (GSEA) was performed on preranked logFC using the R package fgsea. All predefined pathways such as GO gene ontology, Hallmark gene sets, and KEGG pathways were downloaded directly from the Molecular Signatures Database (MSigDB).

### 2.3. RT-qPCR

Total RNA was transcribed into first-strand cDNA using an iScript cDNA Synthesis Kit (Bio-Rad, Hercules, CA, USA). A real-time qPCR reaction was run with PrimeTime Gene Expression Master Mix (IDT, Coralville, IA, USA) using a CFX384 Touch^TM^ Real-Time PCR Detection System (Bio-Rad, Hercules, CA, USA). mRNA expression levels were normalized against *Abelson* (*ABL*), *beta-glucuronidase* (*GUSB*), or *beta-actin* (*ACTB*) expression. Data were expressed as mean ± SEM. Each spot on the graph represents an individual sample. *p*-values for statistical significance were obtained using unpaired Student’s t-test or ANOVA test (Tukey’s multiple comparison). *p* < 0.05 was considered significant.

## 3. Results and Discussion

Pediatric patients were diagnosed with CML at a median of 10 years (interquartile range (IQR): 9–13) compared to 54 years (IQR: 33–62) for adult patients. At diagnosis, pediatric patients had higher platelet counts (*p* = 0.0004) than adult patients. Median WBC counts were 255,000 and 143,000 in pediatric and adult patients, respectively, similar to prior reports [1,2], despite lacking statistical significance (Table 1 and Appendix A).

A total of 1276 genes were differentially expressed in either adult or pediatric CML CD34+ cells compared to healthy CD34+ cells, 174 of which were expressed similarly in pediatric and adult CML CD34+ cells (55 up- and 119 downregulated) (Appendix A). There were 883 differentially expressed genes (376 up- and 507 downregulated) in adult CML CD34+ cells compared to adult healthy CD34+ cells (Appendix A) and 567 differentially expressed genes (207 up- and 360 downregulated) in pediatric CML CD34+ cells compared to pediatric healthy CD34+ cells (Appendix A). At least 92% of CD34+ cells from pediatric and adult CML patients were BCR-ABL+ by fluorescent in situ hybridization (Appendix A), suggesting that gene expression differences between CML and healthy samples reflect true differences between leukemic and healthy CD34+ cells. Moreover, increased *BCR-ABL* expression was verified by RT-qPCR in pediatric and adult CML CD34+ cells (Appendix A).

Interestingly, we observed increased expressions of *GATA1* and *TAL1*, well known as transcriptional regulators for erythrocyte differentiation, in pediatric CML CD34+ cells compared to pediatric healthy CD34+ cells (Appendix A). Pathway analysis showed that several pathways were differentially regulated in pediatric CML compared to healthy controls (Appendix A). As previously reported in CML molecular pathogenesis [4,5,6], we found that Notch/Wnt, CBL, and Rho pathways were differentially expressed in pediatric CML compared to healthy controls (Appendix A). In addition, several genes involved in these pathways, such as *NOTCH1*, *CBL*, *NCOR2*, *TLE1*, and *E2F2*, were significantly decreased in pediatric CML CD34+ cells compared to healthy CD34+ cells (Appendix A).

Next, we directly compared pediatric and adult CML CD34+ cells to identify differentially regulated genes and pathways. Supervised principal component analysis (PCA) showed two distinct populations (Figure 1A), suggesting that the transcriptomes differ between pediatric and adult CML. We observed distinct gene expression profiles across pediatric and adult CML samples (Figure 1B). However, some heterogeneity was noted, with four adult CML samples clustering with pediatric CML samples in certain genes. Two of these patients were young adults at diagnosis (age 23 and 30), while the other two were older (age 55 and 62). WBC counts, platelet counts, and spleen sizes were variable among these four patients and did not explain the differences in gene expression profiles.

A total of 398 genes were differentially expressed (258 up- and 140 downregulated) in pediatric CML CD34+ cells compared to adult CML CD34+ cells (Figure 1C and Appendix A). Several interesting pathways were enriched in pediatric CML (Figure 1D and Appendix A). These differences were not identified when comparing pediatric healthy and adult healthy CD34+ cells using the same stringent conditions (FDR ≤ 0.05). With less stringent conditions (FDR ≤ 0.8), we observed 80 differentially expressed genes (27 up- and 53 downregulated) in pediatric healthy CD34+ cells compared to adult healthy CD34+ cells; however, only one gene (*BEST1*) was differentially expressed in the same direction as in pediatric vs. adult CML CD34+ cells (Appendix A). This suggests that our finding of transcriptomic differences between pediatric CML and adult CML is not attributable to age-related phenomena in healthy cells.

Our pathway analysis showed that the Rho pathways were most significantly downregulated in pediatric CML CD34+ cells compared to adult CML CD34+ cells, excluding a number of inflammation-related pathways (Figure 1D and Appendix A). To examine whether this was disease-related rather than age-related, we performed additional gene set enrichment analysis (GSEA) ranked by log-fold change. Several Rho pathways were upregulated in pediatric healthy CD34+ cells compared to adult healthy CD34+ cells, possibly reflecting differences due to age. Nevertheless, the Rho_GTPASES_ACTIVATE_NADPH_OXIDASES pathway was downregulated in pediatric CML CD34+ cells compared to adult CML CD34+ cells (data not shown). Thus, the Rho pathway was specifically downregulated in pediatric CML, suggesting a unique molecular pathway contributing to pediatric but not adult CML. Several regulator/effector genes in the Rho pathway were differently expressed in pediatric CML CD34+ cells compared to adult CML CD34+ cells (Figure 1E). The Rho family of GTPases is a subfamily of the Ras superfamily that regulates intracellular actin dynamics including organelle development, cytoskeletal dynamics, and cell movement. Rho signaling also affects the interaction between tumor cells, stromal cells, and the extracellular matrix, which may influence disease outcomes [7]. In CML, this pathway is activated through a Dbl homology domain and a Src homology 3 domain of Bcr-Abl [5], resulting in a proliferative advantage and induced abnormal adhesion and migration of cells [8]. Our data suggest that the Rho pathway is less critical for the pathogenesis of pediatric CML compared to adult CML.

To further define the direct comparison of pediatric CML and adult CML, we performed a single-model comparison of pediatric CML vs. adult CML after normalizing with each healthy control: (pediatric CML vs. pediatric healthy) vs. (adult CML vs. adult healthy). Importantly, we recapitulated the results demonstrating that Rho pathways are significantly downregulated in pediatric CML CD34+ cells compared to adult CML CD34+ cells (Appendix A). GSEA analysis showed the same conclusion (Appendix A). Similarly, we observed that several genes in the Rho pathway were differently expressed in pediatric CML CD34+ cells compared to adult CML CD34+ cells (Appendix A).

We selected significantly dysregulated genes in the Rho pathway to verify their gene expressions by RT-qPCR (Figure 1F and Appendix A). As previously reported [9], we used three internal controls including *Abelson* (*ABL*), *beta-glucuronidase* (*GUSB*), and *beta-actin* (*ACTB*) for qPCR. The patterns of gene expression were consistent with RNA-seq data using all internal controls. *VAV2*, a guanine nucleotide exchange factor (GEF) for Rho-family GTPase members and a known oncogene [10], was upregulated in adult CML CD34+ cells seven-fold (*p* = 0.0157) compared to pediatric CML CD34+ cells. *Rho GTPase Activating Protein 27 (ARHGAP27)* was upregulated in adult CML CD34+ cells 3.7-fold (*p* = 0.0453) compared to pediatric CML CD34+ cells. Several genes involved in the regulation of nicotinamide adenine dinucleotide phosphate (NADPH) oxidase, one of the best-characterized Rho GTPase-regulated systems [11], were differently expressed in CML. Compared to healthy CD34+ cells, *Neutrophil Cytosolic Factor 1 (NCF1)*, *Cytochrome B-245 Beta Chain (CYBB)*, and *S100 Calcium-Binding Protein A8 (S100A8)* were highly increased in adult CML CD34+ cells but not in pediatric CML CD34+ cells. Moreover, *NCF1* and *CYBB* were significantly upregulated in adult CML CD34+ cells 2.2-fold (*p* = 0.0458) and 3.26-fold (*p* = 0.006), respectively, compared to pediatric CML CD34+ cells. However, *Deleted in Liver Cancer 1 (DLC1; Rho GTPase activating protein)* was significantly upregulated only in pediatric CML CD34+ cells compared to healthy CD34+ cells, and its expression was 1.74-fold higher (*p* = 0.0044) compared to adult CML CD34+ cells. *DLC1* acts as a tumor suppressor by negatively regulating Rho from active GTP-bound Rho to inactive GDP-bound Rho [12]. While DNA methylation of the *DLC1* promoter is predominantly responsible for its downregulation in hematological malignancies such as acute lymphoblastic leukemia and chronic lymphocytic leukemia [12,13], *DLC1* hypermethylation was found in relatively low frequencies in myeloid leukemia [13]. Our transcriptome data show differential *DLC1* expression in pediatric and adult CML; whether this is due to differences in methylation will be investigated in the future.

## 4. Conclusions

We identified differentially expressed genes and pathways unique to pediatric CML compared to adult CML. Although prior reports have compared molecular signatures between adult CML CD34+ cells and healthy CD34+ cells [14,15], our study is the first to compare transcriptome profiles of CML CD34+ cells across different age groups. Pathway analysis suggests the potential importance of the repressed Rho pathway in pediatric CML CD34+ cells. Furthermore, with new Ras inhibitors under development, our findings have potential clinical implications. Bolouri et al. recently investigated the molecular landscape of pediatric AML and found differences in mutated genes, structural variants, and DNA methylation patterns among patients of different ages [16]. Similarly, our results demonstrate unique molecular characteristics of pediatric CML that may contribute to clinical differences between children and adults with CML. Our study included three young adult patients (diagnosed at age 23, 30, and 33), two of whom had gene expression profiles that were similar to the pediatric samples. Whether the biology of CML in young adults differs from that in older adults and in children will be investigated in the future. A better understanding of the molecular biology of CML across different ages will provide insights into the pathogenesis of pediatric CML and potentially inform future treatment decisions.

## Figures and Tables

**Figure 1 cancers-13-06263-f001:**
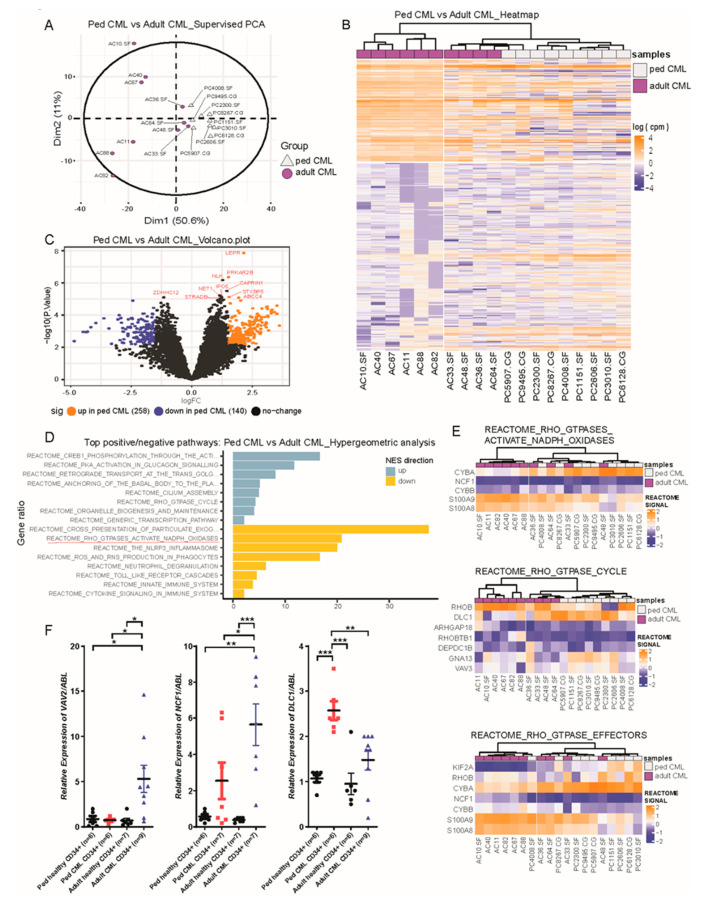
Direct comparison between pediatric and adult CML CD34+ cells shows that genes involved in the Rho pathway are differentially expressed in pediatric CML. (**A**) multi-dimensional scaling (MDS) for supervised PCA was completed with DEG between pediatric and adult CML CD34+ cells. DEG defined as FDR of ≤0.05 and absolute log2 fold-change > 1. (**B**) Heatmap and (**C**) Volcano plot with DEG in pediatric CML CD34+ cells compared to adult CML CD34+ cells. (**D**) Barplot of top REACTOME pathways differentially expressed in pediatric CML CD34+ cells compared to adult CML CD34+ cells. (**E**) Heatmap of the Rho pathways with DEG between pediatric and adult CML CD34+ cells. (**F**) mRNA expression levels of dysregulated genes were examined by RT-qPCR and normalized against *Abelson* (*ABL*) expression. Several genes involved in the Rho pathway were differentially expressed between pediatric and adult CML CD34+ cells. Each spot on the graph represents an individual sample. * *p* < 0.05, ** *p* < 0.01, *** *p* < 0.001.

**Table 1 cancers-13-06263-t001:** Clinical and demographic characteristics of CML patients at diagnosis.

Clinical/Demographic Features	Pediatric (*n* = 9)		Adult (*n* = 10)		Pediatric vs. Adult
	*n*	%	*n*	%	*p*-value *
Demographic characteristics					
Age at diagnosis (years)					
Median (IQR)	10 (9–13)		54 (33–62)		<0.0001
Sex					
Male	8	88.9	9	90.0	1.0000
Female	1	11.1	1	10.0	
Race/Ethnicity					
Asian	4	44.4	1	10.0	0.1771
Hispanic	2	22.2	2	20.0	
White non-Hispanic	2	22.2	6	60.0	
Black non-Hispanic	0	0	1	10.0	
Other	1	11.1	0	0	
Clinical features at diagnosis					
CML diagnosis					
Chronic phase	7	77.8	10	100.0	0.2105
Unknown/not reported	2	22.2	0	0	
WBC count (×10^9^/liter) ^$^					
Median (IQR)	255 (95–351)		143 (64–260)		0.4908
Platelet count (×10^9^/liter) ^$^					
Median (IQR)	627 (617–870)		305 (205–371)		0.0004
Spleen size (cm) ^+^					
Median (IQR)	5 (2–5)		0 (0–0)		0.0739

Abbreviations: IQR, interquartile range; WBC, white blood cell; * *p*-values from Fisher’s exact test (categorical variables) or Wilcoxon rank sum test (continuous variables); ^$^ based on *n* = 7 pediatric patients (all chronic phase) and *n* = 10 adult patients with data available; **^+^** spleen size calculated as centimeters (cm) below the costal margin at diagnosis; based on *n* = 5 pediatric patients (all chronic phase) and *n* = 10 adult patients with data available.

## Data Availability

The RNA sequencing data have been deposited in NCBI’s Gene Expression Omnibus database (GEO accession number GSE163690). The clinical data are available on request from the corresponding author Kathleen M. Sakamoto.

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
