# Peer review of "Comparison of the Transcriptomic Signatures in Pediatric and Adult CML"

_cancers, 2021, doi:10.3390/cancers13246263_

Round 1

Reviewer 1 Report

Dear Authors,

presented short communication article provides detailed insight into differences between adult and pediatric CML CD34+ cell, that could be potentially used in clinics. This research is very interesting and I would suggest it for a publication.

I have only minor note: On the page 3 lines 94 and 101 there is written Rho_GTPASES_ACTIVATE_NADPH_ OXIDASES. It seems out of context only Rho should be there.

So I conclude that after careful spellcheck a checking for typos I suggest this manuscript for the publication.

Author Response

Reviewer #1:

Presented short communication article provides detailed insight into differences between adult and pediatric CML CD34+ cell, that could be potentially used in clinics. This research is very interesting and I would suggest it for a publication.

Response:

We would like to again thank the reviewer for the positive comments and suggestions outlined below.

1.1. Minor:

I have only minor note: On the page 3 lines 94 and 101 there is written Rho_GTPASES_ACTIVATE_NADPH_ OXIDASES. It seems out of context only Rho should be there.

1.1. Response:

Thank you. We have now corrected this in the manuscript.

Reviewer 2 Report

This is a concise and well written manuscript. The methods are clearly explained and the figures are clear and well made.

I have only one issue with this manuscript which is that a key comparison is missing from the analysis. On line 38, the comparisons are described:

“Four comparisons were performed: (1) pediatric CML vs. pediatric healthy, (2) adult CML vs. adult healthy, (3) pediatric CML vs. adult CML, and (4) pediatric healthy vs. adult healthy CD34+ cells”.

The authors do pairwise comparisons and then compare the results from these afterwards using the p-values as cutoffs (e.g. in the venn diagram in Supplementary Figure 1A). This is ok for finding genes that are differentially regulated in both pediatric and adult samples, but not for finding genes that are differentially regulated in pediatric vs adult samples (for example, a gene could have a p-value of 0.05 in the pediatric samples and 0.051 in the adult samples and would be classified as different, when it is in fact very similar).

This missing comparison is: (pediatric CML vs. pediatric healthy) vs (adult CML vs. adult healthy).

The correct way to do this comparison is during the differential expression calculations. The authors use Limma voom for normalisation and this comparison can be done in Limma using a “contrast matrix”, e.g.

cmtx <- makeContrasts("(pediatricCML – pediatricHealthy) – (adultCML - adultHealthy)", levels=design);

v <- voom(data, design);

fit <- lmFit(data, design);

fit <- contrasts.fit(fit, cmtx);

fit <- eBayes(fit);

where data is the normalised counts.

This gives the statistical “difference in differences”, and provides a p-value for each gene. Then all genes where the differential expression is statistically different between pediatric and adult samples (and vice versa) can be found. I think that this comparison would make the manuscript much more complete and should be reasonably easy to do. I’m not entirely convinced by comparing pediatric CML directly to adult CML as age is a confounding factor (the authors do state this) and the comparison I have suggested would clear this up.

Minor points:

Line 79: “Two of these patients were young adults at diagnosis (age 23 and 30)”. In Table 1 the ages are given as 9 -13 and 33 – 62. Please can the authors clarify this.

Can the authors make any comment on if/how the numbers of samples used affect the results.

Author Response

Reviewer #2:

This is a concise and well written manuscript. The methods are clearly explained and the figures are clear and well made.

Response:

We would like to thank the reviewer for the positive comments and helpful suggestions outlined below.

2.1. Major:

I have only one issue with this manuscript which is that a key comparison is missing from the analysis. On line 38, the comparisons are described:

“Four comparisons were performed: (1) pediatric CML vs. pediatric healthy, (2) adult CML vs. adult healthy, (3) pediatric CML vs. adult CML, and (4) pediatric healthy vs. adult healthy CD34+ cells”.

The authors do pairwise comparisons and then compare the results from these afterwards using the p-values as cutoffs (e.g. in the venn diagram in Supplementary Figure 1A). This is ok for finding genes that are differentially regulated in both pediatric and adult samples, but not for finding genes that are differentially regulated in pediatric vs adult samples (for example, a gene could have a p-value of 0.05 in the pediatric samples and 0.051 in the adult samples and would be classified as different, when it is in fact very similar).

This missing comparison is: (pediatric CML vs. pediatric healthy) vs (adult CML vs. adult healthy).

The correct way to do this comparison is during the differential expression calculations. The authors use Limma voom for normalisation and this comparison can be done in Limma using a “contrast matrix”, e.g.

cmtx <- makeContrasts("(pediatricCML – pediatricHealthy) – (adultCML - adultHealthy)", levels=design);

v <- voom(data, design);

fit <- lmFit(data, design);

fit <- contrasts.fit(fit, cmtx);

fit <- eBayes(fit);

where data is the normalised counts.

This gives the statistical “difference in differences”, and provides a p-value for each gene. Then all genes where the differential expression is statistically different between pediatric and adult samples (and vice versa) can be found. I think that this comparison would make the manuscript much more complete and should be reasonably easy to do. I’m not entirely convinced by comparing pediatric CML directly to adult CML as age is a confounding factor (the authors do state this) and the comparison I have suggested would clear this up.

2.1. Response:

Thank you. We broke the analysis up into 3 separate analyses in order to better visualize each comparison separately and for cross-comparison purposes. However, we agree that adding the comparison into the same model, what the reviewer labeled as “difference in differences”, would improve the manuscript and have added this as one of the contrasts in the Limma analysis. Importantly, we recapitulated the results demonstrating that Rho pathways are highly downregulated in pediatric vs adult CML; this was true for both over-representation and gene set enrichment (GSEA) analysis. We have now included this in the revised Supplementary Figure 5 and 6. Moreover, three of the genes that we validated in the original manuscript were also significant (p < .05, FDR of < .12, logFC > 1); this table is now included in the revised Supplementary Table 4.

2.2. Minor:

Line 79: “Two of these patients were young adults at diagnosis (age 23 and 30)”. In Table 1 the ages are given as 9 -13 and 33 – 62. Please can the authors clarify this.

2.2. Response:

Thank you for the comment. Table 1 presents the interquartile range (IQR) for age along with the median age for each patient group; therefore, “9-13” and “33-62” represent the middle 50% of ages for pediatric and adult patients, respectively, rather than the minimum and maximum ages in each group of patients. Complete clinical data including patient ages are shown in Supplementary Table 1.

2.3. Minor:

Can the authors make any comment on if/how the numbers of samples used affect the results?

2.3. Response:

This is an excellent point and we will continue to analyze data from additional patients in the future. However, based on our analysis, we do not expect the conclusion of the manuscript will significantly change.